# Zoonotic *Hantaviridae* with Global Public Health Significance

**DOI:** 10.3390/v15081705

**Published:** 2023-08-08

**Authors:** Rui-Xu Chen, Huan-Yu Gong, Xiu Wang, Ming-Hui Sun, Yu-Fei Ji, Su-Mei Tan, Ji-Ming Chen, Jian-Wei Shao, Ming Liao

**Affiliations:** 1School of Life Science and Engineering, Foshan University, Foshan 528225, China; raycechan@163.com (R.-X.C.); ghy17806245037@163.com (H.-Y.G.); 15922471545@163.com (X.W.); myeohui@sina.com (M.-H.S.); yufei_ji@foxmail.com (Y.-F.J.); meitan001@foxmail.com (S.-M.T.); 2College of Animal Science and Technology, Zhongkai University of Agriculture and Engineering, Guangzhou 510230, China

**Keywords:** *Hantaviridae*, hemorrhagic fever with renal syndrome, hantavirus cardiopulmonary syndrome, taxonomy, evolution, pathogenicity

## Abstract

*Hantaviridae* currently encompasses seven genera and 53 species. Multiple hantaviruses such as Hantaan virus, Seoul virus, Dobrava-Belgrade virus, Puumala virus, Andes virus, and Sin Nombre virus are highly pathogenic to humans. They cause hemorrhagic fever with renal syndrome (HFRS) and hantavirus cardiopulmonary syndrome or hantavirus pulmonary syndrome (HCPS/HPS) in many countries. Some hantaviruses infect wild or domestic animals without causing severe symptoms. Rodents, shrews, and bats are reservoirs of various mammalian hantaviruses. Recent years have witnessed significant advancements in the study of hantaviruses including genomics, taxonomy, evolution, replication, transmission, pathogenicity, control, and patient treatment. Additionally, new hantaviruses infecting bats, rodents, shrews, amphibians, and fish have been identified. This review compiles these advancements to aid researchers and the public in better recognizing this zoonotic virus family with global public health significance.

## 1. Introduction

In 2017, *Bunyaviridae* was officially upgraded to *Bunyavirales*, and the genus *Hantavirus* in *Bunyaviridae* was upgraded to *Hantaviridae* [1]. Multiple hantaviruses such as Hantaan virus (HTNV), Seoul virus (SEOV), Dobrava-Belgrade virus (DOBV), Puumala virus (PUUV), Andes virus (ANDV), and Sin Nombre virus (SNV), are highly pathogenic to humans [2]. They cause hemorrhagic fever with renal syndrome (HFRS) and hantavirus cardiopulmonary syndrome or hantavirus pulmonary syndrome (HCPS/HPS) worldwide (Table 1) [3]. This review aims to consolidate these advancements to help researchers and the public to better understand this zoonotic virus family of global public health importance.

## 2. Diseases Caused by Hantaviruses

Hantaviruses pathogenic to humans have been categorized into Old World hantaviruses (OWHVs) and New World hantaviruses (NWHVs) based on their geographical distribution. OWHVs such as HTNV, SEOV, and DOBV are the primary causative agents of HFRS and are distributed in Asia and Europe. NWHVs including ANDV and SNV are the major etiological agents of HCPS/HPS and are distributed in South America and North America [4]. Globally, about 150,000 to 200,000 HFRS or HCPS/HPS cases are reported annually, with case fatality rates (CFR) ranging from 1 to 15% for HFRS and 30–50% for HCPS/HPS [5].

HFRS was first documented in Korea in 1951 [2]. This disease is usually characterized by five phases, namely, fever, hypotension, oliguria, polyuria, and recovery, typically occurring after an incubation period of 2–3 weeks. Acute kidney injury, a hallmark feature of HFRS, is marked by kidney swelling, proteinuria, and hematuria [6]. Symptoms of HFRS include fever, headache, low back pain, visual impairment, gastrointestinal symptoms (e.g., nausea, vomiting, diarrhea, and bloody stool), and proteinuria [3]. Hemorrhagic symptoms resulting from thrombocytopenia include petechiae of the skin and mucosa, hematuria, hemoptysis, conjunctival congestion, gastrointestinal bleeding, and even intracranial hemorrhage. Disseminated intravascular coagulation (DIC) may also occur in severe HFRS cases. The hypotension phase is characterized by sudden hypotension and shock due to microvascular leakage, which can result in sudden death [5]. The oliguria phase is caused by renal failure, with the symptoms of oliguria (usually <400 mL/day) and proteinuria. During the diuretic phase, renal function gradually recovers, leading to increased urine output. While complete renal function is possible after a long recovery period, chronic renal failure and hypertension may also occur [7]. In addition to the typical renal disease caused by HFRS, some patients may present with extrarenal symptoms such as acute respiratory distress syndrome (ARDS), cholecystitis, pericarditis, and encephalitis [6]. Infection with PUUV in humans can lead to a milder form of HFRS known as nephropathia epidemica (NE) [8].

HCPS/HPS was first documented in North America in 1993 [3]. This disease is typically divided into the prodromal, cardiopulmonary, and recovery phases, occurring after the incubation period of 1–7 weeks [3]. The hallmark features of HCPS/HPS are microvascular leakage and ARDS. The prodromal symptoms include fever, headache, myalgia, nausea, vomiting, and flu-like symptoms (e.g., cough and dyspnea), accompanied by thrombocytopenia [5]. The cardiopulmonary phase lasts for several days and is characterized by tachycardia, arrhythmias, and cardiogenic shock. Pulmonary capillary leakage in this phase can lead to respiratory failure, pulmonary edema, hypotension, bilateral pulmonary infiltration, and pleural effusion. Acute kidney injury and proteinuria also occur in some HCPS/HPS patients. HCPS/HPS can lead to acute deaths during the cardiopulmonary phase [9].

HTNV was first discovered in South Korea in 1978 [10]. HTNV is mainly distributed in China, South Korea, Russia, and Vietnam [5,11]. More than 90% of global HFRS cases are reported from mainland China, with most of them being caused by HTNV [12]. About 9000–12,000 HFRS cases are reported in China each year, with a CFR of approximately 1%. These cases are distributed throughout China [13]. In South Korea, about 400–600 cases of HFRS are reported annually, with a CFR of 1–2% [14].

SEOV was first discovered in South Korea in 1982 [15]. While SEOV mainly circulates in China and South Korea, it is the only hantavirus that is globally distributed [16]. This is consistent with the global distribution of its hosts Norway or brown rats (*Rattus Norvegicus*) [17]. SEOV infection accounts for about 25% of HFRS cases worldwide, with a CFR less than 1% [3].

DOBV was first discovered in Slovenia in 1992 [18]. It is mainly distributed in Slovenia, Czech, Poland, Russia, Serbia, Greece, Germany, Denmark, France, and other European countries [19]. The pathogenicity of DOBV varies depending on its genotypes. The Dobrava and Sochi genotypes are highly pathogenic, with a CFR of 10–15% [20]. The Kurkino genotype is less virulent and causes only mild HFRS, with a CFR of 0.3–0.9%. The Saaremaa genotype does not cause mortality [19]. 

PUUV was first discovered in Finland in 1980 [21]. PUUV is mainly distributed in central, northern, and eastern European countries [5]. It caused more than 3000 HFRS cases each year in Europe between 2010 and 2020. These are usually mild, with a CFR of 0.1–0.4% [3,5].

ANDV was first discovered in Argentina in 1995 [22]. ANDV is mainly distributed in Argentina and Chile and causes dozens of HCPS/HPS cases annually with a CFR of about 40%. ANDV can be transmitted among humans [23]. 

SNV was first detected in 1993 in the USA [24]. SNV circulates in the USA, Canada, and Mexico, and causes dozens of HCPS/HPS cases annually with a CFR of 30–35% [25,26].

Other viruses in *Orthohantavirus* such as Tula virus (TULV) in many Eurasian countries [27], Bayou virus (BAYV) in the USA [28], Choclo virus (CHOV) in Panama [29], and *Laguna Negra virus* (LANV) in South America [30] have been identified as pathogenic to humans.

## 3. Morphology and Genomics of Hantaviruses

Hantavirus virions are usually spherical with a quasi-symmetric icosahedral lattice (T = 12). The diameter of the virions is 120–160 nm [31]. The lipid bilayer of the viral envelope is about 5 nm thick. The surface spikes embedded in the viral envelope are composed of the glycoproteins Gn and Gc. Each spike is formed by a tetramer of Gn and Gc (Figure 1) extending from the lipid bilayer to about 10 nm. Inside the envelope are the nucleocapsids composed of many copies of the nucleocapsid (N) protein, which interact with the three segments of the viral genome and the viral RNA-dependent RNA polymerase (RdRp) [32].

Like most members of *Bunyavirales*, hantaviruses have a genome encompassing large (L), medium (M), and small (S) single-stranded negative-sense RNA (–ssRNA) segments, consisting of about 10,000–15,000 nucleotides in total (Figure 2) [33]. The L segment encodes RdRp, which mediates the transcription and replication of viral genomic RNA. The M segment encodes a glycoprotein precursor (GPC) that is co-translated and cleaved by host cell signal peptidases to form Gn and Gc [34]. Gn and Gc bind to cell receptors, regulate immune responses, and induce protective antibodies [4]. The S segment encodes the N protein, which binds to and protects the viral RNA molecules. The S segment of some orthohantaviruses such as ANDV, PUUV, SNV, TULV, and LANV also encodes a nonstructural protein (NS) that inhibits the interferon production in host cells [35].

The terminal untranslated region (UTR) sequences of the genomic segments of hantaviruses are highly conserved and complementary. They form stalk structures and participate in the replication and transcription of viral genomic RNA (vRNA) [36,37].

**Figure 2 viruses-15-01705-f002:**
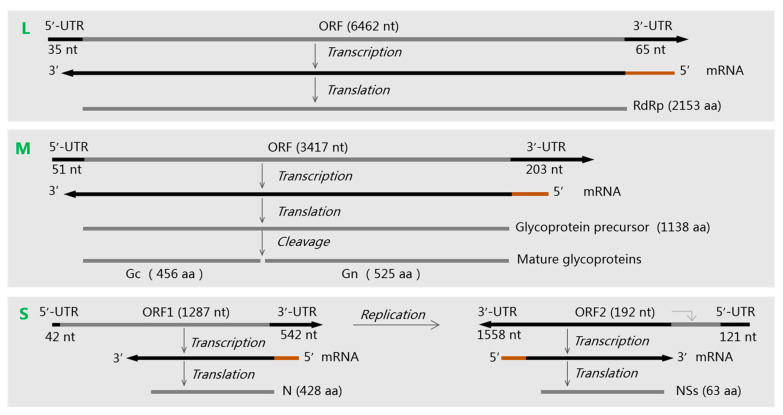
The three genomic RNA segments (L, M, and S) of an Andes virus and their encoding proteins. Their sequence accession numbers in GenBank are NC003468, NC003467, and NC003466 [38]. Brown lines represent the 5′-methylated cap and some nucleotides transferred from the host cell mRNAs to the viral mRNAs. ORF: open reading frame. UTR: untranslated region.

## 4. Taxonomy and Distribution of Hantaviruses

The *Hantaviridae* Study Group of the International Committee on Taxonomy of Viruses (ICTV) designated the method DEmARC to classify new hantavirus genera and species. DEmARC uses the phylogenetic relationships and pairwise evolutionary distance (PED) values of the concatenated S and M segment sequences, calculated with the maximum likelihood method and the Whelan and Goldman (WAG) substitution model. The PED cutoff value of 0.1 is used for species demarcation in *Hantaviridae* [39].

In 2018, only one virus genus (*Orthohantavirus*) was classified in *Hantaviridae*. In 2019, three new virus genera (*Loanvirus*, *Mobatvirus*, and *Thottimvirus*) in *Hantaviridae* were classified. In 2023, four subfamilies (*Actantavirinae*, *Agantavirinae*, *Mammantavirinae*, and *Repantavirinae*), seven genera (*Actinovirus*, *Agnathovirus*, *Loanvirus*, *Mobatvirus*, *Orthohantavirus*, *Thottimvirus*, and *Reptillovirus*), and 53 species in *Hantaviridae* were classified (Table 2 and Figure 3) [40,41]. The species *Orthohantavirus robinaense* in *Orthohantavirus* (Table 2), which was reported in Australia in 2017, should belong to the *Mobatvirus* genus (Figure 3) [42]. These taxa were consistent with the phylogenetic relationships in their S-genomic sequences (Figure 3) [43], and were largely consistent with their host distribution and conserved sequences at the genome termini [39]. The ICTV *Hantaviridae* Study Group will reclassify in 2023–2024 and only evaluate those viruses with complete genome sequences [44].

The genus *Actinovirus* covers five species of fish viruses, with four identified in China in 2018 and one identified in Europe in 2020 [45]. The genus *Agnathovirus* covers one species of fish virus identified in China in 2018 [45]. The genus *Loanvirus* covers two species of bat viruses identified in China in 2013 and the Czech Republic in 2015, respectively [46,47]. The genus *Mobatvirus* covers three species of bat viruses identified in Vietnam, China, and the Philippines, respectively, in 2015–2016 [48,49,50], besides one mole virus identified in Poland in 2014 and one shrew virus identified in Russia in 2019 [51,52]. The current genus *Orthohantavirus* covers 38 species with hosts in rodents, moles, shrews, and humans [40]. As above-mentioned, HTNV, SEOV, DOBV, PUUV, SNV, and ANDV are widely distributed and highly pathogenic to humans [2]. The genus *Thottimvirus* covers two species of shrew viruses identified in India in 2007 and in Japan in 2008 [53,54]. The genus *Reptillovirus* covers one species of gecko viruses identified in China in 2018 [45].

In 2023, the names of hantavirid species were markedly changed. For example, the species *Hantaan orthohantavirus* was renamed as *Orthohantavirus hantanense* (Table 2).

Some species of hantaviruses have been classified into several lineages. For example, the HTNVs isolated from China were classified into 10 lineages as per their S or M genomic segment sequences [55]. SEOVs were classified into six lineages (1–6) as per their M genomic segment sequences. Lineages 1, 2, 3, and 5 of the SEOVs were isolated from China, while lineage 4 included isolates from China, South Korea, Japan, Singapore, Vietnam, and the USA. Lineage 6 SEOVs were isolated from the U.K., except for one strain from Brazil [56].

## 5. Evolution of Hantaviruses

Numerically, the most common events in the genomic evolution of hantaviruses are single-nucleotide substitutions. Nucleotide insertions and deletions including those leading to frameshift mutations are important in the evolution of hantaviruses, particularly in the macroevolution (e.g., inter-species evolution) of hantaviruses [57].

Like other viruses with multiple genomic segments, hantaviruses can utilize genomic recombination and reassortment to rapidly change their genomic sequences. Genomic reassortment and recombination events occur frequently in nature and have been demonstrated in in vitro experiments. These events have facilitated the adaptation of hantaviruses to multiple hosts and ecosystems [4,58].

Recombination events in the S or M genomic segment in OWHVs such as HTNV, SEOV, ANDV, TULV, and PUUV have been recorded [55,58]. A recombinant HTNV strain, A16, isolated from Shaanxi Province in China, had a parent from a hantavirid species other than HTNV [55], suggesting that recombinant events in hantaviruses can occur among different hantavirid species.

Reassortment events of NWHVs (e.g., SNV) in rodents in the USA have been recorded. These events mainly exchange the viral S or M segments [4,58]. ANDV-SNV reassortment events and HTNV-SEOV reassortment events were experimentally confirmed [58]. A total of 19.1% of PUUV genomes obtained from rodents in 2005–2009 were identified as reassortants [4].

The evolution of hantaviruses was likely driven by host adaptation and geographical isolation [59]. Pathogenic hantaviruses have been isolated from various rodent species. For instance, HTNV has been isolated from *Apodemus agrarius*, SEOV from *Rattus norvegicus*, PUUV from *Myodes glareolus*, ANDV from *Oligoryzomys longicaudatus*, and SNV from *Peromyscus maniculatus*. It has been long believed that rodents are the original hosts of various hantaviruses and that one hantavirus species circulates largely in one rodent species [2]. Although recent studies suggest that one hantavirus species can circulate in a few rodent species (Table 2), the genera and species of *Hantaviridae* are largely consistent with their host species, and orthohantaviruses have evolved into different species in the hosts of the *Murinae*, *Arvicolinae*, *Neotominae*, and *Sigmodontinae* subfamilies [59]. Meanwhile, HTNV and SEOV have formed multiple lineages corresponding to their geographic distribution [56].

Multiple studies have shown that mammalian hantaviruses may originate from hantaviruses in shrews, bats, or moles [46], which might come from reptile or fish hantaviruses [45].

## 6. Replication of Hantaviruses

The replication processes of hantaviruses in host cells include attachment to host cells and internalization, membrane fusion, transport and release into the cytoplasm, transcription, replication, translation of the genome, and the assembly and release of the virion (Figure 4) [60]. Vascular endothelial cells and macrophages are the primary sites of hantavirus replication [61]. 

The attachment and entry of hantaviruses into host cells are mediated by the binding of viral glycoproteins to host cell surface receptors [61]. Some hantaviruses attach and enter host cells through integrins (e.g., αVβ3 and αVβ1), which are transmembrane heterodimer glycoproteins composed of α and β subunits [60]. In addition to integrins, decay acceleration factors (e.g., DAF/CD55) and complement receptors (e.g., gC1qR/P32) have also been proposed as candidate cell attachment factors for some hantaviruses [60]. Protocadherin-1 likely plays a role in the attachment and entry of all NWHVs [62]. After attachment to cell surface receptors, hantaviruses rely on several pathways for entry including macropinocytosis and endocytosis, which are either clathrin-, calveolin-, or cholesterol-dependent [60]. For instance, HTNV and SEOV are internalized by clathrin-mediated endocytosis. ANDV can be internalized through cholesterol-mediated micropinocytosis [60].

After internalization, virions form clathrin-coated vesicles, which can be formed with clathrin-coated cell membranes [61]. Virions are then transported to the early endosomes, and eventually to the late endosomes and lysosomal compartments. The membrane fusion of the viral envelope with the endosome is accompanied by a decrease in pH value, from the weak acidity of the early endosome to the strong acidity of the late endosome. Hantaviruses require acidity (pH 5.8–6.3) for membrane fusion [63]. Gc is a type II viral fusion protein. The conformational change of Gc is triggered by endosomal acidification, and then the Gc fusion peptide is inserted into the endosomal membrane, leading to further conformational change, which mediates the fusion of the viral envelope and endosomal membrane [64].

The fusion of the viral envelope and endosomal membrane promotes the release of the viral ribonucleoproteins (RNP) complex into the cytoplasm, initiating the transcription and replication of the viral genome [31]. RdRp mediates the transcription and replication of the viral genome, and the replication of the hantavirus genome occurs in the cytoplasm [7]. vRNA replicates into complementary RNA (cRNA), which serves as a template for vRNA replication. Genome replication starts from scratch and proceeds through forward cRNA, which replicates out vRNA and is encapsulated by N proteins to form RNPs [61].

N proteins play a variety of roles in the viral replication cycle. They participate in transcription together with viral RdRp to promote mRNA translation [31]. RdRp initiates transcription, which produces the viral mRNAs. The N-terminal of the L protein has endonuclease activity, which can cut and utilize capped primers from the mRNA of host cells to synthesize viral mRNAs (cap-snapping) [31]. The viral mRNAs transcribed from the L and S genomic segments are translated on episomal ribosomes, and the viral mRNAs transcribed from the M genomic segments in membrane-bound ribosomes are translated in the endoplasmic reticulum (ER) and rough endoplasmic reticulum (RER) [65].

The viral GPC is cleaved in the ER into the Gn and Gc glycoproteins at the conserved pentapeptide motif WAASA [31], which are further glycosylated in the ER and transported to the Golgi complex, where they form heterodimers [66]. Gn and Gc are modified by N-glycan chains, which stabilize the spike structure and play a key role in the virion assembly [64]. All virions are assembled in the Golgi apparatus, and the newly synthesized virions bud into the Golgi pool, during which the cytoplasmic tail of Gn interacts with the RNP complex [60]. The virions are then transported to the cell membrane, where they are released by exocytosis. Hantaviruses are usually assembled at the Golgi apparatus, and some hantaviruses could be assembled at the plasma membrane through the fusion of viral vesicles and cell membranes [31].

The host mechanisms of innate and acquired immunity in mammals are crucial to inhibiting hantavirus replication [67].

## 7. Transmission of Hantaviruses

Although hantaviruses can be inactivated by heating at 60 °C for 30 min, organic solvents, hypochlorite solvents, or ultraviolet light [5], they are relatively stable in the external environment. They can survive for 10 days at room temperature and more than 18 days at low temperatures (e.g., 4 °C) [12]. This facilitates the transmission of hantaviruses.

Unlike the fact that various bunyaviruses (e.g., Rift Valley fever virus) are transmitted by insects, human hantaviruses are mainly transmitted by rodents [45,54,68]. Hantaviruses usually cause asymptomatic and persistent infections in rodents, except that Syrian hamsters infected with ANDV can show typical symptoms of HCPS/HPS. Rodents, shrews, moles, and bats are the reservoirs of some hantaviruses, which can transmit the virus horizontally and vertically (Figure 5). 

Infected animals can spread hantaviruses to other sensitive animals through aerosols or droplets formed from their excreta or secretions (feces, urine, and saliva), through the consumption of contaminated food, or through biting and scratching [69,70]. They can also transmit the hantavirus through the fecal–oral route, and females can transmit the virus through the placenta [71]. Mating and fighting among sensitive animals also aid in virus transmission [72].

Rodents can also transmit hantaviruses to humans through aerosols or droplets formed from their excreta and through the consumption of contaminated food [3]. Rodent bites and scratches are important for virus transmission to humans [73]. The prevalence rate of hantaviruses (e.g., SEOV) is up to 80% among traders who cultivate and breed rodents and those who keep pet rodents (e.g., rats and mice) [74,75]. An outbreak of SEOV in North America with 31 confirmed cases resulted from contact with pet rats [73].

ANDV is currently the only known hantavirus that can be transmitted from human to human. It usually occurs after close contact with an infected person [76]. ANDV may be secreted into human saliva and transmitted through the respiratory tract via airborne droplets released by coughing or sneezing [77]. Studies have shown that ANDV can also be transmitted to newborns through breast milk, and transmission between humans can also occur through the digestive tract or placenta [78].

Investigations have shown that HFRS can be transmitted to animals or humans through the bites of mites [79,80]. Investigations have also shown that pigs can be infected with HTNV without severe symptoms [81] and sows can vertically transmit the virus through the placenta [82]. A swine breeder was infected with hantavirus, but had no history of mite bites or rodent infestation in the living environment, so pig-to-human transmission of the virus could be possible [83]. However, the relevant investigations were neglected, possibly because they were published in Chinese.

HTNV and PUUV antibodies have been detected in cattle, deer, and rabbits. SNV antibodies have been detected in cats and dogs [84]. The roles of these mammalian hosts in the ecology of hantaviruses remain unknown.

## 8. Pathogenesis of Hantaviruses

Human infections with hantaviruses can cause acute diseases, with the severity varying depending on the viral species and strains [3,19,20]. HFRS primarily affects the kidney and blood vessels, while HCPS/HPS mainly affects the heart and lungs [67]. Other organs and systems such as the nervous system, spleen, and liver can also be affected [85].

Hantavirus infections in both animals and humans mainly occur in renal or pulmonary endothelial cells (ECs) and macrophages [65]. The main pathogenesis of HFRS and HCPS/HPS covers increased vascular permeability and acute thrombocytopenia [5]. The infection begins with the interaction between hantavirus glycoproteins (Gn and Gc) and β-integrin receptors (e.g., αvβ1 and αvβ3) on target cell membranes [86]. Immature dendritic cells located near ECs transport virions from lymphatic vessels to local lymph nodes to infect more ECs. These cells act as antigen-presenting cells and induce immune reactions, especially those associated with macrophages and CD8^+^ T lymphocytes [5]. Delayed type I interferon responses lead to higher virus titers [87]. After immune activation, cytotoxic T lymphocytes produce pro-inflammatory cytokines that can damage the infected ECs, leading to increased vascular permeability and inflammatory reactions [88]. Activation of complements and the release of pro-inflammatory cytokines such as interferon (IFN), interleukins (IL-1, IL-6, and IL-10), and tumor necrosis factor-α (TNF-α) play a crucial role in changing EC permeability [86]. Different hantaviruses elicit different immune responses. High levels of cytotoxic CD8^+^ T lymphocytes can be detected in HFRS patients. Elevated IL-6 levels are associated with thrombocytopenia and renal failure [86]. 

Human leukocyte antigen (HLA) is responsible for presenting viral antigens to T lymphocytes. Genetic susceptibility to severe HFRS disease is related to the HLA type, and different hantaviruses are associated with different HLA haplotypes [65].

Natural reservoir hosts of hantaviruses often exhibit asymptomatic and persistent infection, which may result from the innate immunity of the reservoir hosts [85]. In rodents, it is possible that T lymphocytes can be regulated with the immune receptor NLRC3, leading to persistent viral infection without symptoms, and the lack of such an immune regulation in humans leads to disease [89]. 

The lack of obvious symptoms in natural hosts and the lack of suitable animal models have limited research on the pathogenesis of HTNV and SEOV [90]. However, studies using *Nlrc3*^−/−^ mice infected with HTNV have shown clinical symptoms and pathological changes resembling those seen in patients with HFRS, suggesting a new model for studying the pathogenesis of HTNV. Monkeys can be infected with PUUV through laboratory infection with similar symptoms to NE and can be used as an animal model of NE [85]. Research into the pathogenesis of HCPS/HPS has been aided by the hamster model, which mimics the pulmonary capillary leak and the hypotension characteristic of human HCPS/HPS [5]. 

## 9. Diagnosis and Treatment of Hantavirus Infections

Epidemiological information and clinical symptoms are useful for the initial diagnosis of hantavirus infections. Confirmation of hantavirus infections should be based on clinical symptoms, epidemiological information, and the detection of the nucleic acids, proteins, and/or antibodies specific to the associated hantavirus [65]. 

No specific antivirals have been approved for HFRS and HCPS/HPS in the USA or Europe [5]. However, some antivirals including ribavirin, fapilavir, and lactoferrin have displayed varying efficacy in the treatment of hantavirus infections [91].

Prophylactic administration of ribavirin and fapilavir in early infection or post-exposure has shown some efficacy [92]. Both ribavirin and fapilavir have shown anti-hantavirus activity in vivo and in vitro [5,91]. Ribavirin has been proven to be effective in the early treatment of HFRS with some limitations including toxicity to humans and animals at high doses and the potential to cause hemolytic anemia [92]. Fapilavir is superior to ribavirin without the side effect associated with anemia [77]. Fapilavir was evaluated in an ANDV/SNV-infected hamster model, where oral administration twice daily of 100 mg/kg significantly reduced the viral RNA load in the blood and the antigen load in the lung. Oral administration of fapilavir before the onset of viremia prevented HCPS, whereas delayed administration had no protective effect [91].

Antibodies include monoclonal and polyclonal antibodies [91]. Monoclonal antibodies that are effective against hantaviruses have been demonstrated in hamster models and sucking mouse models [91]. Monoclonal antibodies have been proven to be effective for the treatment of HCPS/HPS induced by ANDV infection in hamster models [93]. Polyclonal antibodies produced by DNA-vaccinated bovines have also displayed protective effects in animal models [65].

Supportive treatment can be used to maintain fluid and electrolyte balance in HFRS patients, while blood transfusions can control bleeding and improve clotting in DIC cases. For HCPS/HPS patients, oxygen supplementation is usually required, and extracorporeal membrane oxygenation may be employed [5].

## 10. Prevention and Control of Hantaviruses

The prevention and control of hantaviruses pathogenic to humans rely on education and actions to block the viral transmission and protect humans.

Epidemiological surveillance and analysis play key roles in identifying risky regions and factors that contribute to hantavirus transmission. To mitigate this risk, rodenticides, traps, and cats can be employed for the targeted control of rodents. Additionally, cleaning and disinfecting human living environments can prevent the contamination of food and other items with hantaviruses from rodent excreta or secretions. 

People at high risk of infection with hantaviruses such as those working in agriculture, forestry, or animal husbandry should take protective measures including wearing masks or face coverings [94]. No vaccines for hantaviruses have been approved worldwide, with the exception of the Republic of Korea (ROK) and mainland China. In the ROK, an inactivated vaccine (Hantavax) has been marketed since 1990 to prevent HFRS caused by HTNV and SEOV, which requires a three- or four-dose regimen [95]. A similar bivalent inactivated vaccine was marketed in 1994 in mainland China [96]. Approximately 2 million doses of HFRS inactivated bivalent vaccine are used annually in mainland China, with vaccine-induced immunity lasting up to 33 months [97].

In recent years, multiple types of new vaccines for hantaviruses have been investigated in laboratories. For instance, a vaccinia virus recombinant vector vaccine expressing the glycoproteins Gn and Gc of HTNV elicited neutralizing antibodies against both HTNV and SEOV in hamster models, although the vaccine had no cross-protection against PUUV [98]. Various DNA vaccines expressing the genomic glycoproteins of OWHVs and NWHVs have also been investigated in hamster models [91].

## 11. Discussion

From the panorama of hantaviruses depicted above, hantaviruses circulate in various mammals (e.g., bats, rodents, shrews, and moles), amphibians, and fish worldwide, and some hantaviruses are highly pathogenic to humans. They usually infect humans through animal-to-human transmission. In recent years, the taxonomy of *Hantaviridae* has been greatly expanded and revised, and research targeting various aspects of this important virus family has made significant progress.

In the years to come, numerous research advances in *Hantaviridae* can be anticipated. Novel genera or species within this family will be discovered, particularly from novel regions or host species. For instance, the novel hantaviruses Rusne virus from the root vole in Lithuania and the Academ virus from moles in Russia have recently been reported [99,100]. Some genomic segments of important known hantaviruses (e.g., Lena virus and El Moro Canyon virus) will be sequenced and analyzed. The structures and functions of more proteins of hantaviruses will be revealed. More evolutionary and ecological features of hantaviruses including the roles of pigs and mites in hantavirus transmission will be further explored, and the phylogenetic distribution and pathogenesis of more hantaviruses will be clarified. More vaccines (e.g., mRNA vaccines) and other measures to control highly pathogenic hantaviruses will be evaluated. The panorama of hantaviruses depicted in this review can be a valuable reference for these explorations.

## Figures and Tables

**Figure 1 viruses-15-01705-f001:**
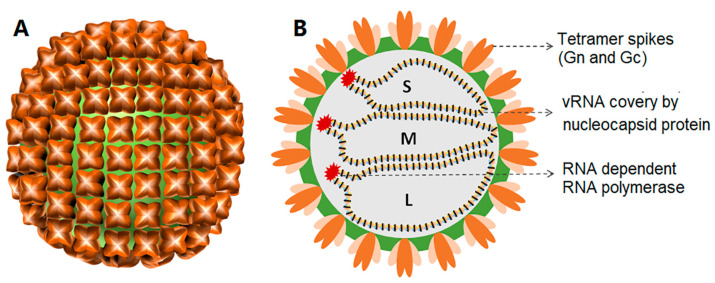
The external (**A**) and interior (**B**) structures of hantaviruses with three genomic RNA segments (L, M, and S).

**Figure 3 viruses-15-01705-f003:**
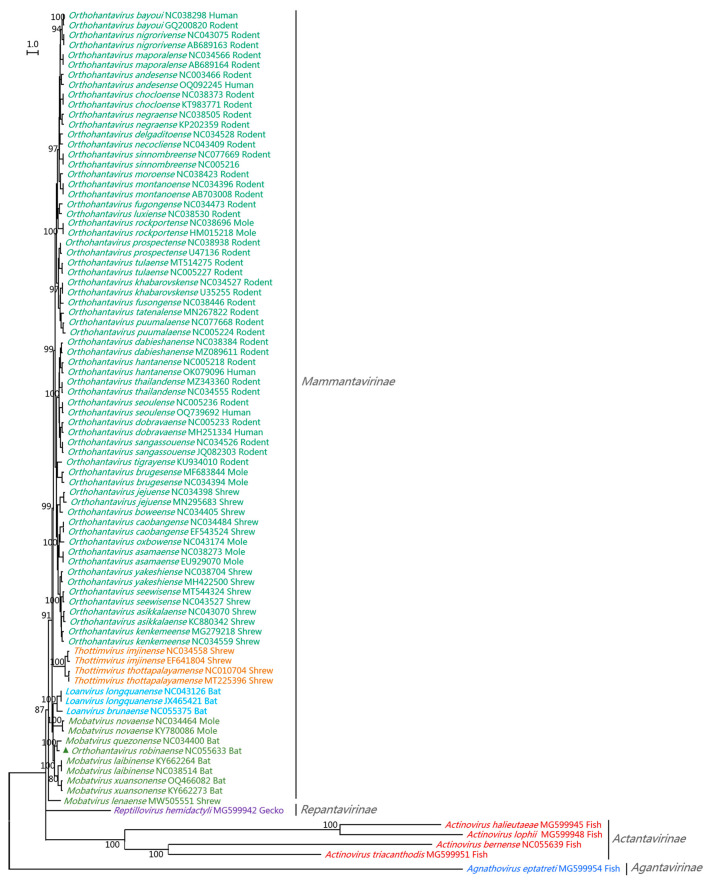
Phylogenetic relationships among the four subfamilies, seven genera, and 53 species in *Hantaviridae*. The relationships were calculated using the nucleotide sequences of the viral S-genomic segments, the software MEGA 11.0, the maximum likelihood method, and the GTR + G + I model. Bootstrap values were calculated using 1000 replicates. Different genera are shown using different colors, and species names are followed with the relevant GenBank accession numbers and hosts. The species *Orthohantavirus robinaense* marked with a triangle should be assigned to the *Mobatvirus* genus rather this phylogenetic tree and its host.

**Figure 4 viruses-15-01705-f004:**
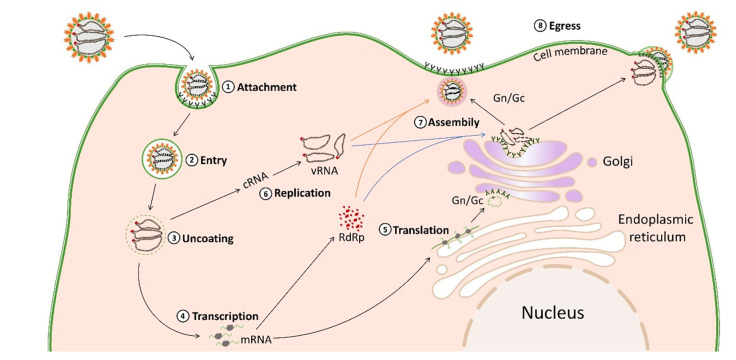
Schematic overview of the replication of some hantaviruses.

**Figure 5 viruses-15-01705-f005:**
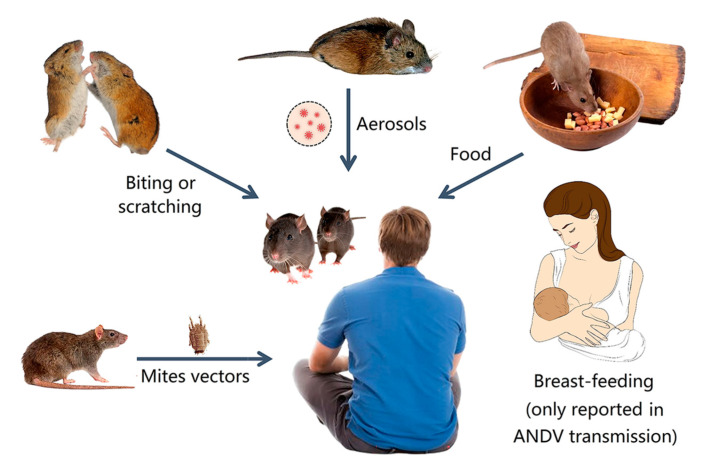
Transmission routes of some hantaviruses in animals and humans.

**Table 1 viruses-15-01705-t001:** Human diseases caused by hantaviruses.

Disease	Affected Organs	Etiological Viruses *	Case Fatality Rate	Distribution
Hemorrhagic fever with renal syndrome (HFRS)	Kidney, spleen, liver, brain, blood vessels, eyes	HTNV	~1%	China, South Korea, Russia, Vietnam
		SEOV	<1%	Worldwide
		DOBV	0–15%	Europe
		TULV	~0%	Eurasia
		PUUV	0.1–0.4%	Europe
Hantavirus cardiopulmonary syndrome or hantavirus pulmonary syndrome (HCPS/HPS)	Lung, heart, spleen, liver, blood vessels	ANDV	~40%	Argentina, Chile
		SNV	30–50%	USA, Canada
		CHOV	12–15%	Panama
		LNV	12–15%	Argentina, Brazil, Paraguay, Bolivia

*: HTNV: Hantaan virus; SEOV: Seoul virus; DOBV: Dobrava-Belgrade virus; TULV: Tula virus; PUUV: Puumala virus; ANDV: Andes virus; SNV: Sin Nombre virus; CHOV: Choclo virus; LNV: Laguna Negra virus.

**Table 2 viruses-15-01705-t002:** The current taxonomy of *Hantaviridae*.

Genus	Species	Hosts
*Actinovirus*	*Actinovirus halieutaeae*	Fish (*Halieutaea stellata*) in China
	*Actinovirus lophii*	Fish (*Lophius litulon*) in China
	*Actinovirus bernense*	Fish (*Perca fluviatilis*) in Switzerland
	*Actinovirus triacanthodis*	Fish (*Triacanthodes anomalus*) in China
*Agnathovirus*	*Agnathovirus eptatreti*	Fish (*Eptatretus burger*) in China
*Loanvirus*	*Loanvirus brunaense*	Bats (*Nyctalus noctule*) in the Czech Republic
	*Loanvirus longquanense*	Bats (e.g., *Rhinolophus sinicus*, *Rhinolophus monoceros*, and *Rhinolophus sinicus)* in China
*Mobatvirus*	*Mobatvirus laibinense*	Bats (*Taphozous melanopogon*) in China and Myanmar
	*Mobatvirus lenaense*	Shrews (*Sorex caecutiens*) in Russia
	*Mobatvirus novaense*	Moles (*Talpa europaea*) in Europe
	*Mobatvirus quezonense*	Bats (*Rousettus amplexicaudatus*) in Philippines
	*Mobatvirus xuansonense*	Bats (*Hipposideros cineraceus*) in Asia
*Orthohantavirus*	*Orthohantavirus asamaense*	Rodents (e.g., *Oligoryzomys longicaudatus*, *Oligoryzomys nigripes*, and *Akodon azarae*), bats (e.g., *Carollia perspicillata* and *Desmodus rotundus*), and humans in South America
	*Orthohantavirus asikkalaense*	Moles (*Urotrichus talpoides*) in Japan
	*Orthohantavirus bayoui **	Shrews (*Sorex minutus*) in Europe
	*Orthohantavirus nigrorivense **	Rodents (*Oryzomys palustris*) and humans in the USA
	*Orthohantavirus boweense*	Rodents (*Sigmodon hispidus*) in the USA
	*Orthohantavirus brugesense*	Shrews (*Crocidura douceti*) in Guinea
	*Orthohantavirus delgaditoense*	Moles (*Talpa europaea*) in Europe
	*Orthohantavirus caobangense*	Rodents (*Sigmodon alstoni*) in Venezuela
	*Orthohantavirus chocloense **	Shrews (*Anourosorex squamipes*) in China and Vietnam
	*Orthohantavirus dabieshanense*	Rodents (*Oligoryzomys fulvescens*) and humans in Panama
	*Orthohantavirus dobravaense **	Rodents (*Niviventer confucianus*) in China
	*Orthohantavirus moroense*	Rodents (e.g., *Apodemus flavicollis*, *Rattus norvegicus*, and *Mus musculus*) and humans in Europe
	*Orthohantavirus fugongense*	Rodents (*Reithrodontomys megalotis*) in Mexico
	*Orthohantavirus fusongense*	Rodents (*Eothenomys eleusis*) in China
	*Orthohantavirus hantanense **	Rodents (*Microtus fortis*) in China
	*Orthohantavirus jejuense*	Rodents (e.g., *Apodemus agrarius*, *Rattus tanezumi*, and *Myospalax psilurus*) and humans in Asia
	*Orthohantavirus kenkemeense*	Shrews (*Crocidura shantungensis*) in South Korea
	*Orthohantavirus khabarovskense*	Shrews (*Sorex roboratus*) in China and Russia
	*Orthohantavirus negraense **	Rodents (*Microtus maximowiczii*) in China and Russia
	*Orthohantavirus luxiense*	Rodents (e.g., *Calomys laucha*, *Calomys callidus*, and *Akodon simulator*) in South America
	*Orthohantavirus maporalense*	Rodents (*Eothenomys miletus*) in China
	*Orthohantavirus montanoense*	Rodents (*Oligoryzomys fulvescens*) in Venezuela
	*Orthohantavirus necocliense*	Rodents (*Peromyscus beatae*) in Mexico
	*Orthohantavirus oxbowense*	Rodents (*Zygodontomys brevicauda*) in Colombia
	*Orthohantavirus prospectense*	Moles (*Neurotrichus gibbsii*) in the USA
	*Orthohantavirus puumalaense **	Rodents (*Microtus pennsylvanicus*) in the USA
	*Orthohantavirus robinaense * ^#^	Rodents (*Myodes glareolus*) and humans in Russia and Germany
	*Orthohantavirus rockportense*	Bats (*Pteropus alecto*) in Australia
	*Orthohantavirus sangassouense*	Moles (*Scalopus aquaticus*) in the USA
	*Orthohantavirus seewisense*	Rodents (e.g., *Hylomyscus simus* and *Hylomyscus endorobae*) in Guinea and Kenya
	*Orthohantavirus seoulense **	Rodents (e.g., *Apodemus flavicollis*, *Apodemus ilex*, and *Apodemus chevrieri*) and shrew (e.g., *Sorex Araneus*, *Sorex caecutiens*, and *Anourosorex squamipes*) in Europe
	*Orthohantavirus sinnombreense **	Rodents (e.g., *Rattus norvegicus*, *Rattus tanezumi*, and *Rattus nitidus*), shrew (e.g., *Suncus murinus*, *Sorex Araneus*, and *Crocidura lasiura*), and humans worldwide
	*Orthohantavirus tatenalense*	Rodents (e.g., *Peromyscus maniculatus*, *Mus musculus*, and *Tamias minimus*) and humans in North America
	*Orthohantavirus thailandense*	Rodents (*Microtus agrestis*) in the UK
	*Orthohantavirus tigrayense*	Rodents (e.g., *Rattus rattus*, *Eliurus majori*, and *Bandicota indica*) and humans in Madagascar, Sri Lanka, and Thailand
	*Orthohantavirus tulaense **	Rodents (*Stenocephalemys albipes*) in Ethiopia
	*Orthohantavirus yakeshiense*	Rodents (*Microtus arvalis*) in Europe
	*Thottimvirus imjinense*	Shrews (*Sorex isodon*) in China and Russia
*Thottimvirus*	*Thottimvirus thottapalayamense*	Shrews (e.g., *Crocidura lasiura*, and *Crocidura shantungensis*) in China and South Korea
	*Reptillovirus hemidactyli*	Shrews (*Suncus murinus*) in Asia
*Reptillovirus*	*Actinovirus halieutaeae*	Geckos (*Hemidactylus bowringii*) in China

*: These viruses are pathogenic to humans. ^#^: This species should be classified into the *Mobatvirus* genus.

## Data Availability

The data that support the findings of this study are available from the corresponding author Ji-Ming Chen upon reasonable request.

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
