# Peer review of "Zoonotic Hantaviridae with Global Public Health Significance"

_viruses, 2023, doi:10.3390/v15081705_

Round 1

Reviewer 1 Report

Norway or brown rat 

Table S1. It would be helpful to indicate which of these viruses is a known human pathogen perhaps with an asterisk * and footnote defining the *.

In treatment, supplemental oxygen may be needed and in very severe cases of pulmonary involvement, extracorporeal membrane oxygenation may be employed.

Some minor copy editing would be helpful to select the best English words and phrases.

Author Response

We would like to express our sincere gratitude to you for taking the time to review our manuscript. Your suggestions have been highly professional and constructive, and we have incorporated all of them into our revised manuscript.

(1) Norway or brown rat.

Response: Accepted and the revision was given in the last line of Page 2.

(2) Table S1. It would be helpful to indicate which of these viruses is a known human pathogen perhaps with an asterisk * and footnote defining the *.

Response: Accepted and revised the table.

(3) In treatment, supplemental oxygen may be needed and in very severe cases of pulmonary involvement, extracorporeal membrane oxygenation may be employed.

Response: Accepted and the revision was given in the last sentence of Section 9.

(4) Some minor copy editing would be helpful to select the best English words and phrases.

Response: Accepted and the manuscript has been polished for better language usage.

Reviewer 2 Report

The review ’Zoonotic Hantaviridae with global public health significance’ by Shao Jian-Wei and co-authors has systematically reviewed the current knowledge within the hantavirus field, especially regarding the host range of this virus, zoonotic transmission, and zoonotic diseases. The manuscript is well-written and I have added a few comments:

1.     Figure 1b. Nucleocapsid (N) protein.

2.     Figure 3. Remove the bootstrap value lower than 75 in the tree.

3.     Figure 5. Change ‘sucking’ to ‘breast-feeding (only reported in ANDV transmission)’.

4.     In Table 2, please highlight the zoonotic viruses, which can cause human diseases in Table 1.

Author Response

We would like to express our sincere gratitude to you for taking the time to review our manuscript. Your suggestions have been highly professional and constructive, and we have incorporated all of them into our revised manuscript.

(1) The review ’Zoonotic Hantaviridae with global public health significance’ by Shao Jian-Wei and co-authors has systematically reviewed the current knowledge within the hantavirus field, especially regarding the host range of this virus, zoonotic transmission, and zoonotic diseases. The manuscript is well-written and I have added a few comments:

Response: Many thanks.

(2) Figure 1b. Nucleocapsid (N) protein.

Response: Accepted and the revision was given in the figure.

(3) Figure 3. Remove the bootstrap value lower than 75 in the tree.

Response: Accepted and the revision was given in the figure.

(4) Figure 5. Change ‘sucking’ to ‘breast-feeding (only reported in ANDV transmission)’.

Response: Accepted and the revision was given in the figure.

(5) In Table 2, please highlight the zoonotic viruses, which can cause human diseases in Table 1.

Response: Accepted and the revision was given in Table 2.